# Human acute inflammatory recovery is defined by co-regulatory dynamics of white blood cell and platelet populations

Brody H. Foy[1,2,3], Thoralf M. Sundt[4], Jonathan C. T. Carlson [1,5] ✉,
Aaron D. Aguirre [1,6,7,8] ✉ & John M. Higgins [1,2,3] ✉

Inflammation is the physiologic reaction to cellular and tissue damage caused by trauma, ischemia, infection, and other pathologic conditions. Elevation of white blood cell count (WBC) and altered levels of other acute phase reactants are cardinal signs of inflammation, but the dynamics of these changes and their resolution are not well established. Here we studied inflammatory recovery from trauma, ischemia, and infection by tracking longitudinal dynamics of clinical laboratory measurements in hospitalized patients. We identified a universal recovery trajectory defined by exponential WBC decay and delayed linear growth of platelet count (PLT). Co-regulation of WBC-PLT dynamics is a fundamental mechanism of acute inflammatory recovery and provides a generic approach for identifying high-risk patients: 32x relative risk (RR) of adverse outcomes for cardiac surgery, 9x RR of death from COVID-19, 9x RR of death from sepsis, and 5x RR of death from myocardial infarction.

Trauma, infection, ischemia, and other pathologic conditions trigger an inflammatory response, a set of orchestrated cellular and tissue-level mechanisms that seek to prevent further injury and to repair existing damage[1,2]. The acute phase of inflammation is typically induced by exogenous molecules from pathogens or endogenous molecules activated by tissue stress or damage[1]. White blood cells (WBCs) resident in damaged tissue and platelets (PLTs) aggregating and activating at sites of vascular injury are key mediators of the downstream response[2,3]. Homeostatic setpoints for blood cell populations are altered temporarily during an acute inflammatory event before returning to baseline during recovery[1,4,5]. While the identities of many molecular inducers and cellular mechanisms have been established, response dynamics at the level of WBC and PLT effector populations are poorly understood. Leukocytosis (elevated WBC count) is a cardinal sign of acute inflammation, but simple elevation in

WBC count is highly non-specific, and the rates of change and resolution in WBC and PLT associated with favorable acute inflammatory responses are not well-defined[1,5]. Patient responses appear to vary dramatically with no clearly defined signs of good prognosis. This fragmented understanding of inflammatory responses at the cellular population level often limits clinical practice to binarized assessments of acute phase reactants and heuristic interpretation of blood counts[5,6].

We hypothesized that successful inflammatory recoveries from diverse pathologic conditions would share some common core dynamics. We therefore analyzed multivariate temporal relationships in clinical laboratory studies from patients responding to multiple types of infection, ischemia, and trauma associated with surgery. Patients with good recoveries showed a consistent underlying response pattern involving co-regulation of WBC and PLT populations:

[1]Center for Systems Biology, Massachusetts General Hospital, Harvard Medical School, Boston, MA, USA. [2]Department of Pathology, Massachusetts General Hospital, Harvard Medical School, Boston, MA, USA. [3]Department of Systems Biology, Harvard Medical School, Boston, MA, USA. [4]Division of Cardiac Surgery, Corrigan Minehan Heart Center, Massachusetts General Hospital, Boston, MA, USA. [5]Cancer Center, Massachusetts General Hospital, Harvard Medical School, Boston, MA, USA. [6]Cardiology Division, Corrigan Minehan Heart Center, Massachusetts General Hospital, Harvard Medical School, Boston, MA, USA. [7]Wellman Center for Photomedicine, Massachusetts General Hospital, Harvard Medical School, Boston, MA, USA. [8]Healthcare Transformation Lab, Massachusetts General Hospital, Harvard Medical School, Boston, MA, USA. ✉e-mail: carlson.jonathan@mgh.harvard.edu; Aguirre.Aaron@mgh.harvard.edu; john_higgins@hms.harvard.edu

uncomplicated recoveries were characterized by exponential decay from a maximum WBC followed by delayed linear growth of PLT. This robust pattern may provide a generic benchmark for inflammatory recovery, analogous to the growth curves used to monitor healthy pediatric development[7].

## Results

### Multivariate test result trajectories during inflammatory recovery

We first considered the inflammatory response to non-emergency cardiac surgery because it involves (1) a significant inflammatory challenge (sternotomy with cardiopulmonary bypass) with well-defined timing and consistent magnitude and (2) relatively stable baseline patient conditions across the cohort. We studied 4693 patients at Massachusetts General Hospital (MGH): an exploratory cohort with surgery between 01-01-2016 and 30-09-2018, $N = 3168$, and a validation cohort with surgery between 01-10-2018 and 31-12-2019, $N = 1525$ (Table 1). Adverse outcomes were defined as death or a post-operative hospital length of stay (LOS) >14 days. Unsupervised clustering of the high-dimensional trajectories (Fig. 1a) of complete blood count (CBC) and basic metabolic panel (BMP) measurements identified high-risk and low-risk subsets, with 22-fold mortality risk differences (0.8–17.8%) (Supplementary Table 1, Supplementary Fig. 1). Multiple individual markers, including WBC and PLT, were significantly auto-correlated and cross-correlated (Supplementary Figs. 2–3), suggesting robust co-regulation of blood cell populations and serum markers, and implying that reduced-dimension phenotypes would retain sufficient information to identify common features of favorable responses.

### WBC-PLT trajectory during inflammatory recovery from cardiac surgery

To reduce dimensionality and retain interpretability[8–12], we performed phase plane analysis, a powerful approach commonly used in engineering and applied mathematics to understand the quantitative behavior of dynamical systems by analyzing subsets of system variables as a function of time[13]. We focused on the WBC-PLT plane because WBCs and PLTs play primary roles in inflammation[1,2,4], and the WBC-PLT combination provided at least as strong statistical association with outcomes as other pairwise combinations (Fig. 1b, Supplementary Tables 2–3). Patient WBC-PLT trajectories show that cardiac surgery is a dramatic inflammatory perturbation, moving more than 95% of patients outside the reference intervals for WBC or PLT (Fig. 1c). To investigate whether cardiac surgery patients who recover well follow homogeneous or heterogeneous paths, we analyzed WBC-PLT trajectories for patients who survived and had LOS < 14 days[14,15]. Individual patient trajectories reflect divergent baseline characteristics and significant variation in clinical events, especially over short periods of time (Fig. 1a), but the shape of the average WBC-PLT recovery trajectory was nevertheless conserved and robust to variation in patient age, LOS, hospital of service, baseline WBC and PLT, standardized pre-operative risk assessment[14,15], operation type, patient gender, year of surgery, and more (Fig. 1c, Supplementary Fig. 4, Supplementary Movie 1). Patients with poor outcomes did not demonstrate this WBC-PLT trajectory shape (Supplementary Fig. 5). Thus, WBC and PLT populations were on average co-regulated in a consistent way during the resolution phase of an effective inflammatory response to cardiac surgery. In a separate validation cohort, patients whose trajectory diverged from the average benchmark had a 14x increased risk of death (CI 8.0–24.1, 0.7–10.2%) on day 3 after surgery and a 22x increased risk (CI: 11.7–39.8, 0.8–17.4%) on day 5 (Fig. 1d).

The persistence of this WBC-PLT trajectory shape motivated investigation of cohorts undergoing six other major surgeries: Cesarean section, colectomy, hip arthroplasty, hysterectomy, Whipple

**Table 1 | Characteristics of the inflammatory cohorts**

| Basic characteristics | Cardiac surgery | Cesarean section | Colectomy | Hip arthroplasty | Hysterectomy | Whipple prodecure | Limb amputation | COVID-19 | C. difficile colitis | Sepsis | Myocardial infarction | Stroke |
|---|---|---|---|---|---|---|---|---|---|---|---|---|
| No. of patients | 4693 | 1273 | 1584 | 3249 | 1049 | 912 | 753 | 1396 | 383 | 4730 | 6240 | 2494 |
| Age: mean (SD) yrs | 63.9 (13) | 33.7 (5.4) | 61.8 (15.2) | 71 (12.4) | 60.3 (12.4) | 67.5 (11.4) | 63.2 (14) | 61.8 (17.5) | 69 (17.1) | 66.8 (16.8) | 70 (13.3) | 70.6 (14.6) |
| Gender: No. male (%) | 3346 (71.3) | 0 (0) | 778 (49.1) | 1177 (36.2) | 0 (0) | 477 (52.3) | 502 (66.7) | 813 (58.2) | 216 (56.4) | 2669 (56.4) | 3963 (63.5) | 1311 (52.6) |
| **Pre-operative (or at admission) tests** | | | | | | | | | | | | |
| White blood cell count: mean (SD) $10^3$/µL | 7.9 (3.1) | 12 (4.7) | 10.8 (6.9) | 9 (6.3) | 7.8 (4.7) | 7.3 (4.2) | 11.7 (6) | 8 (4.5) | 14.6 (14.8) | 14.2 (10.3) | 10.5 (7.1) | 10.1 (6.3) |
| Platelet count: mean (SD) $10^3$/µL | 217.9 (71.7) | 204 (91) | 275 (143) | 230 (105) | 280 (141) | 232 (109) | 308 (149) | 218 (100) | 278 (160) | 232.3 (132.2) | 235 (92) | 239 (102) |
| Red cell distribution width: mean (SD) % | 14.2 (2.2) | 14.3 (1.7) | 15.5 (2.8) | 14.3 (1.9) | 16.2 (3.4) | 14.9 (2.3) | 15.9 (2.5) | 14.1 (2.2) | 15.8 (2.9) | 15.6 (2.6) | 14.3 (2) | 14.5 (2.3) |
| Hematocrit: mean (SD) % | 38.1 (6.1) | 34.3 (4.3) | 33.8 (6.9) | 34.6 (5.3) | 33.7 (5.7) | 34.9 (5.5) | 29.1 (4.9) | 38.8 (6.4) | 34.7 (6.2) | 34.7 (7.0) | 38.2 (6.8) | 37.5 (6.9) |
| **Clinical outcomes** | | | | | | | | | | | | |
| Inpatient stay: mean (SD) days | 8.8 (10.4) | 3.5 (2.8) | 8.2 (8.9) | 4 (3.4) | 4.7 (3.8) | 7.2 (7.3) | 10.3 (11.3) | 11.7 (12.9) | 10.5 (12.2) | 10.2 (10.5) | 8.7 (9.3) | 10.4 (11.4) |
| 30-day mortality: No. (%) | 107 (2.3) | 2 (0.2%) | 122 (7.7%) | 48 (1.5%) | 15 (1.4%) | 24 (2.6%) | 80 (10.6%) | 189 (13.5%) | 45 (11.7%) | 508 (10.7%) | 579 (9.3%) | 320 (12.8%) |

SD standard deviation.

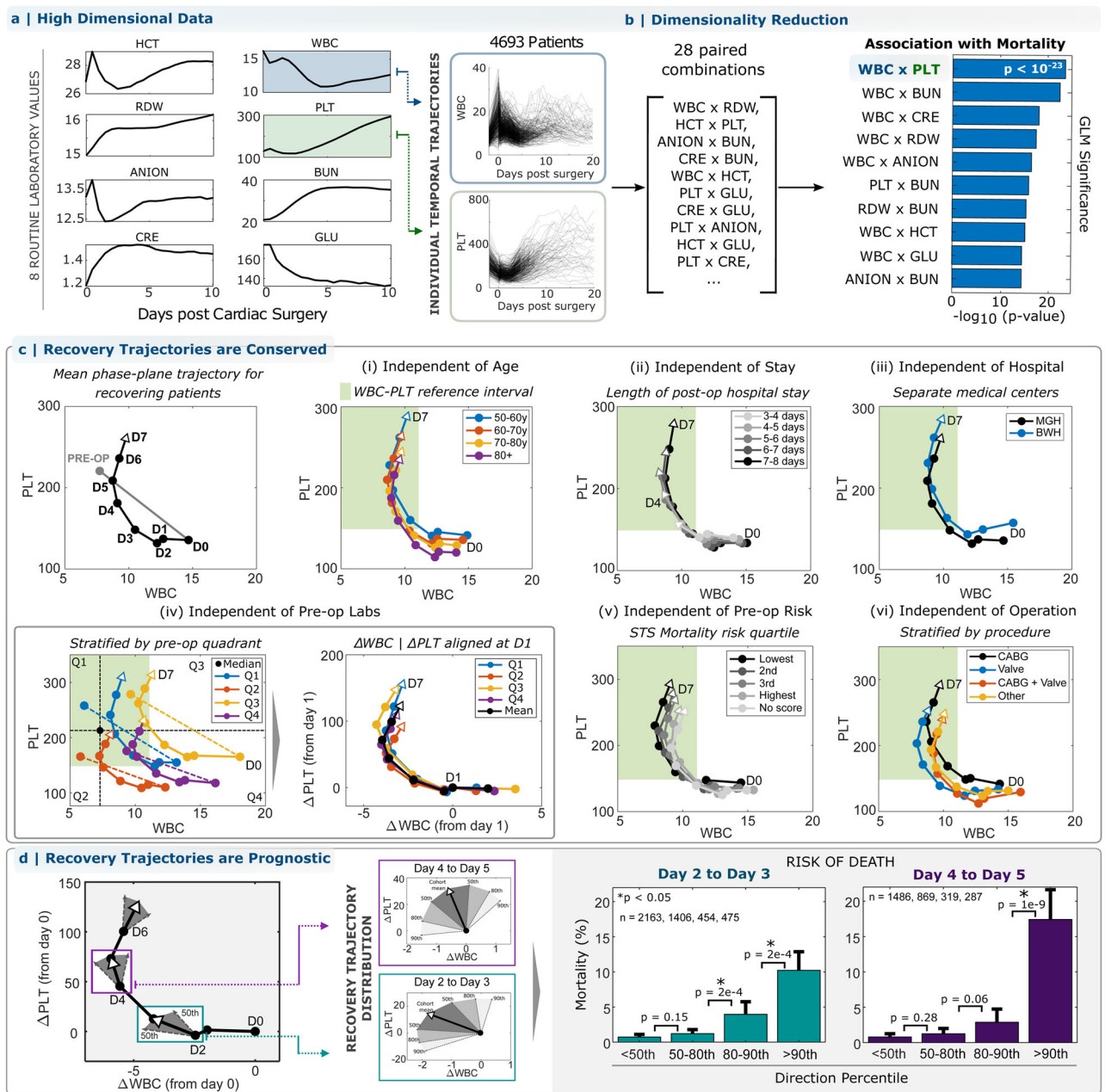

**Fig. 1 | Analysis of the acute inflammatory response to non-emergency cardiac surgery in the WBC-PLT phase plane reveals a consistent shape of recovery.** **a** High-dimensional trajectories of routine cellular and serum markers following cardiac surgery are associated with patient outcome. **b** Trajectories retain association in reduced dimensions, with WBC-PLT as the 2D combination most strongly associated with outcomes. **c** The average patient with a good recovery (surviving, LOS < 14 days) follows a consistent recovery trajectory in the WBC-PLT phase-plane, which persists independent of (i) age, (ii) length of stay, (iii) medical center, (iv) pre-op marker values, (v) pre-op risk defined by the STS risk score[15], (vi) surgery sub-type, and gender and surgery year (Supplementary Fig. 4). **d** Deviation from the mean recovery trajectory is associated with adverse outcomes in retrospective analysis. The mean trajectory along with the 50th, 80th, and 90th percentiles for daily directional changes are shown. Directional deviation from the mean trajectory

is associated with significant (star: *$p < 0.05$) increased mean mortality for patients above the 90th percentile compared to those below the 50th: 14x (CI 8.0–24.1, 0.7–10.2%, two-sided chi-squared test, $\chi^2$: 150, $p = 1e$–16, df = 1) on day 3 after surgery and 22x (CI: 11.7–39.8, 0.8–17.4%, two-sided chi-squared test, $\chi^2$: 177, $p = 1e$–16, df = 1) on day 5 (error bars denote 95% CI on the mean). Circles in panels (**d**) denote 1-day time intervals, or from pre-op to immediately post-op. See Supplementary Fig. 12 for trajectories for alternate test result pairs. Trajectories for WBC lineages (neutrophils, lymphocytes, etc.) are shown in Supplementary Fig. 14. See Supplementary Movie 1 for a video animation of recovery trajectories from (**c**), iv. Source data are provided as a Source data file. HCT: hematocrit, WBC: white blood cell count, RDW: red cell distribution width, PLT: platelet count, ANION: anion gap, BUN: blood-urea nitrogen, CRE: creatinine, GLU: glucose, MGH: Massachusetts General Hospital, BWH: Brigham and Women's Hospital.

procedure, and limb amputation (Table 1). After adjusting for baseline WBC and PLT and the magnitude of initial WBC response, Fig. 2a shows that the average favorable WBC-PLT recovery trajectories of all 7 cohorts are similar. This average WBC-PLT recovery shape is robust to the large differences in demographics, baseline health, and surgical indication across these cohorts (Supplementary Fig. 6).

## WBC-PLT recovery shows exponential WBC decay and linear PLT growth

Across the 7 surgical cohorts, WBC recovery dynamics could be approximated as an exponential decay from the maximum post-op WBC ($WBC_{max}$) toward the patient's pre-inflammation baseline or homeostatic setpoint ($WBC_{setpoint}$) with a decay constant of $k_{WBC_{decay}}$

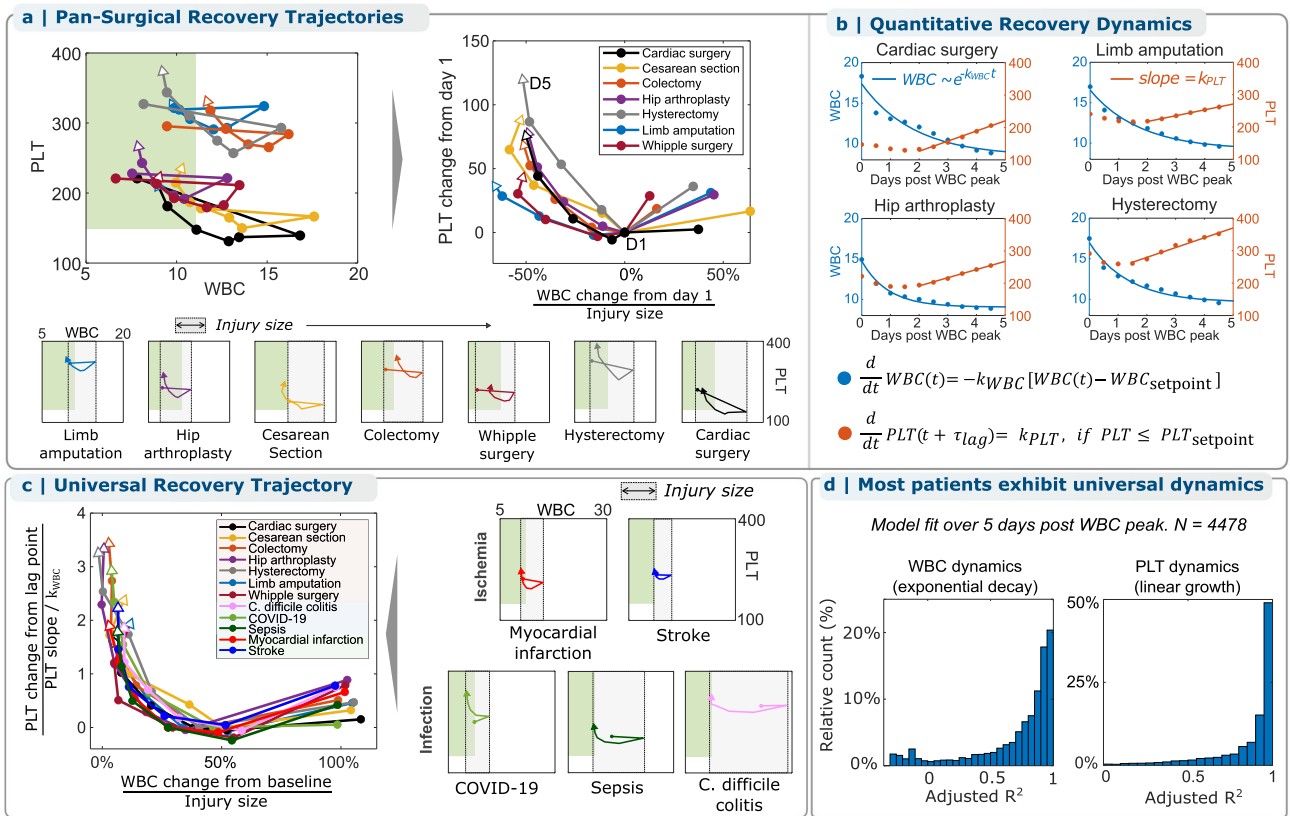

**Fig. 2 | The WBC-PLT recovery shape is conserved across multiple types of traumatic, infectious, and ischemic inflammatory stimuli and demonstrates exponential WBC decay and linear PLT growth. a** Six additional types of surgery are associated with a similar average shape of WBC-PLT recovery. **b** WBC dynamics are well-approximated as simple exponential decay from a maximum WBC, and PLT dynamics as linear growth starting $\tau_{PLT_{delay}}$ days after the maximum WBC. **c** Infectious and ischemic inflammatory stimuli demonstrate the same average WBC-PLT recovery shape, and a non-dimensionalized[17] model shows that physiologic responses to all studied inflammatory stimuli share the same characteristic features. **d** Most patients with good outcomes follow trajectories well-approximated by this model, with median model fits (adjusted $R^2$) of 0.85 and 0.95 for WBCs and PLTs. See Supplementary Fig. 15 for model fits for the other 8 cohorts and Supplementary Fig. 7 for fitted model parameters ($k_{WBC}, k_{PLT}$), for all cohorts. Source data are provided as a Source data file. WBC: white blood cell count, PLT: platelet count.

days$^{-1}$ (Fig. 2b):

$$\text{WBC}(t) = \left(\text{WBC}_{max} - \text{WBC}_{setpoint}\right) e^{-k_{WBC_{decay}} \cdot (t - t_{WBC_{max}})}$$

for $t \geq t_{WBC_{max}}$, where $\text{WBC}_{max} = \text{WBC}(t_{WBC_{max}})$.

Starting $\tau_{PLT\_delay}$ days after $t_{WBC_{max}}$, PLT counts grew approximately linearly with a slope $k_{PLT_{growth}} \frac{10^3}{\mu l \cdot day}$ (Fig. 2b):

$$\text{PLT}(t) = \text{PLT}\left(t_{WBC_{max}} + \tau_{PLT_{delay}}\right) + k_{PLT_{growth}} \cdot \left(t - t_{WBC_{max}} - \tau_{PLT_{delay}}\right)$$

$$\text{for } t \geq \left(t_{WBC_{max}} + \tau_{PLT_{delay}}\right).$$

A simple mechanistic model consistent with these observed recovery dynamics is one of a homeostatic feedback control system[4,16] where (i) excess WBCs are removed at a constant rate per WBC, with negligible additional WBCs introduced into the circulation, (ii) PLTs are produced at a constant rate with negligible consumption starting ~2 days later, and (iii) both processes continue at least until the patient's baseline setpoints are reached. Model fits of $k_{WBC_{decay}}$ across the cohorts (Supplementary Fig. 7) suggest WBCs are removed at random after circulating for an average duration of ~1–2 days after $t_{WBC_{max}}$, corresponding to a half-life of 0.7–1.4 days. PLT growth starts about 2 days after the peak WBC, continuing until

reaching or exceeding the setpoint. In the cardiac surgery cohort where more data was available, most recovering patients appeared to overshoot their baseline $\text{PLT}_{setpoint}$ (Fig. 1c). WBC population dynamics were driven primarily by neutrophils (Supplementary Fig. 14).

### WBC-PLT recovery trajectory in infection and ischemia

Given the resilience of the conserved recovery trajectory shape across a diverse range of surgical trauma, we investigated whether the pattern persisted in non-traumatic inflammatory conditions: three types of infection (COVID-19, *Clostridium difficile* colitis, and sepsis) and two types of ischemia (myocardial infarction and stroke). Figure 2c shows that the same basic response shape is identified and that WBC and PLT kinetics for the average recovering patient in each cohort are well-explained by exponential WBC decay and delayed linear PLT growth. As in the surgical cohorts, this characteristic response is preserved when the cohorts are stratified by gender, age, LOS, and baseline WBC-PLT (Supplementary Fig. 8). To reveal characteristic features of WBC and PLT dynamics intrinsic to all inflammatory recoveries, we adjusted for the differences in baseline and rate parameters by non-dimensionalizing[17] the model using the following dimensionless variables $(w, p, t^*)$:

$$w = \frac{\text{WBC} - \text{WBC}_{setpoint}}{\text{WBC}_{max} - \text{WBC}_{setpoint}}$$

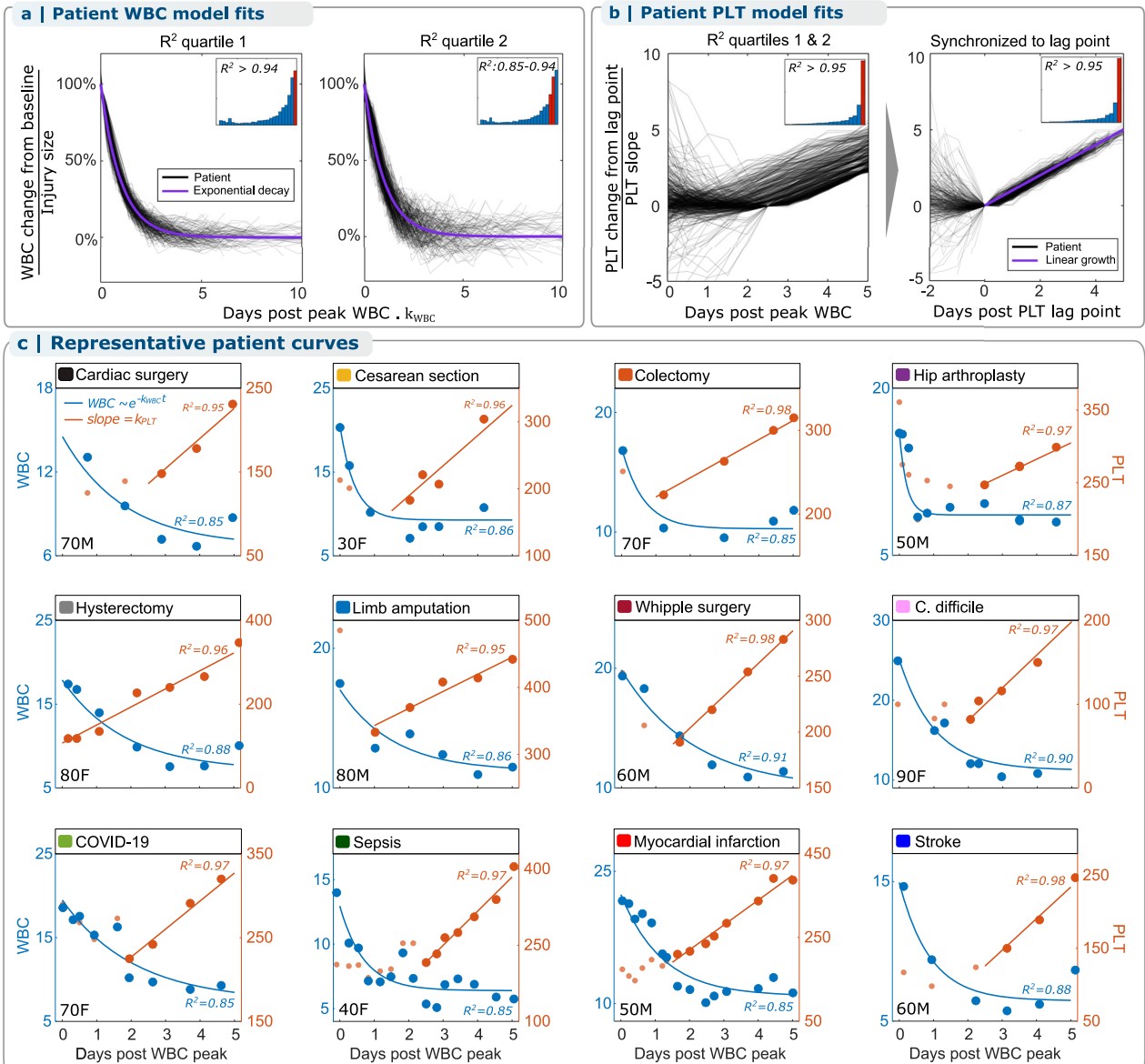

**Fig. 3 | Most individual recovering patients follow the same WBC-PLT recovery shape, independent of inflammatory stimulus. a** Across all 12 cohorts, individual patient WBC recovery dynamics are well-approximated by the model with median adjusted $R^2 = 0.85$, as are individual PLT recovery dynamics (**b**) with a median adjusted $R^2$ of 0.95 following a variable delay. **c** WBC-PLT dynamics for representative patients follow the model. Representative patients are those with each inflammatory condition whose WBC-PLT data had model fits closest to the cohort median adjusted $R^2$. These patients exhibit exponential WBC decay and delayed linear PLT increase during recovery despite significant differences in gender and age (bottom left of each panel), inflammatory stimulus, CBC frequency, and magnitude of WBC and PLT count changes. See Supplementary Figs. 9–10 for additional representative WBC-PLT dynamics for patients whose raw WBC-PLT data had model fits near the 25th and 75th percentiles of the cohort adjusted $R^2$ distributions. Thin black lines in panels (**a**), (**b**) are shown for 500 patients randomly selected from the adjusted $R^2$ distributions. Source data are provided as a Source data file. WBC: white blood cell count, PLT: platelet count, M: male, F: female.

$$p = \frac{\mathrm{PLT} - \mathrm{PLT}\left(t_{\mathrm{WBC}_{max}} + \tau_{\mathrm{PLT}_{delay}}\right)}{\left(\dfrac{k_{\mathrm{PLT}_{growth}}}{k_{\mathrm{WBC}_{decay}}}\right)}$$

$$t^* = \left(t - t_{\mathrm{WBC}_{max}}\right) \cdot k_{\mathrm{WBC}_{decay}}, \text{for } t \geq t_{\mathrm{WBC}_{max}}$$

This variable transformation (Fig. 2c) makes clear that WBC-PLT dynamics during recovery from all of these inflammatory states is well-characterized as exponential WBC decay and delayed linear PLT

growth:

$$w(t^*) = e^{-t^*}, \, t^* \geq 0$$

$$p(t^*) = t^* - \tau_{\mathrm{PLT}_{delay}} \cdot k_{\mathrm{WBC}_{decay}}, \, t^* \geq \tau_{\mathrm{PLT}_{delay}} \cdot k_{\mathrm{WBC}_{decay}}$$

For all patients with sufficient data across all inflammatory cohorts, this model fit individual patient WBC-PLT trajectories with a median adjusted $R^2$ of 0.84 for WBC and 0.92 for PLT (Fig. 2d). Figure 3 shows that typical recovering patients tend to follow this recovery shape, regardless of age or condition, with additional individual patient trajectories provided in Supplementary Figs. 9–10.

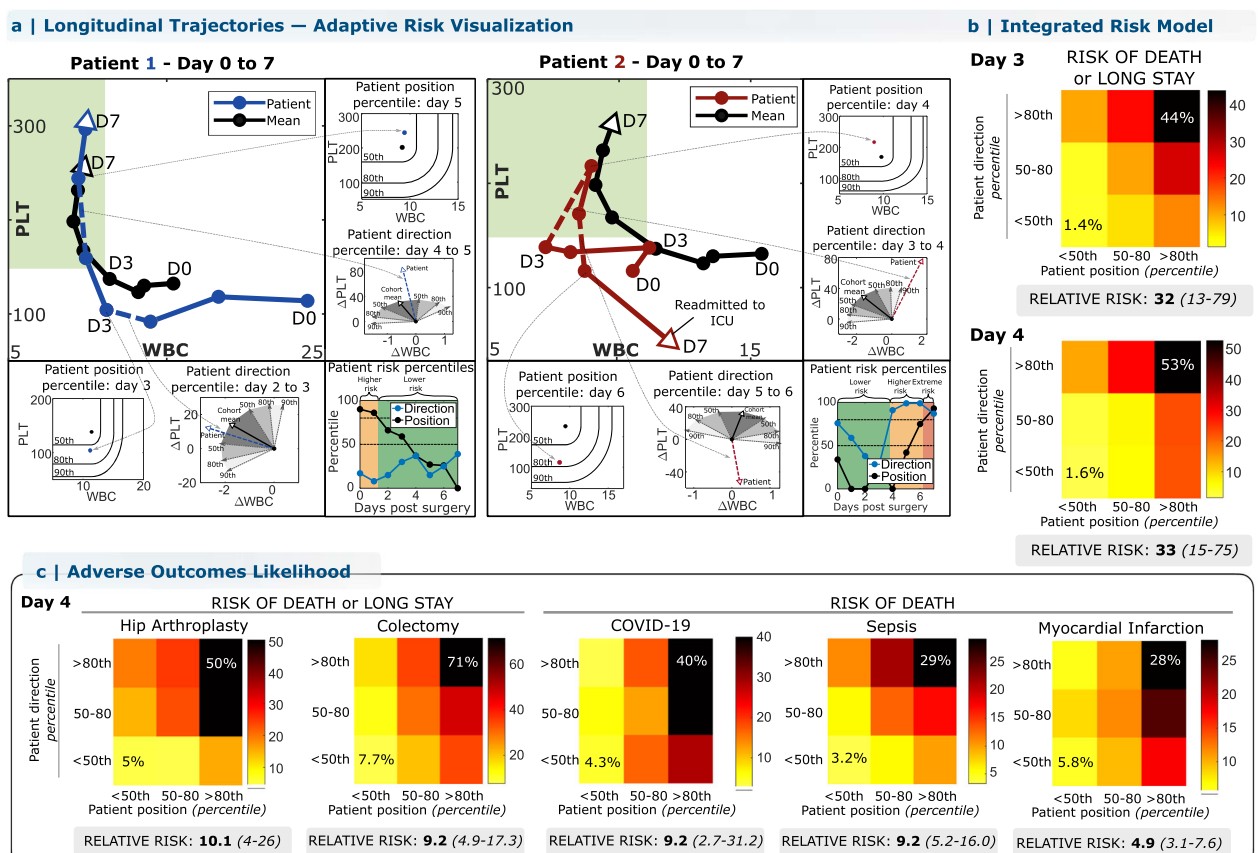

**Fig. 4 | The conserved WBC-PLT trajectory shape provides a generic benchmark for personalized patient risk assessment.** WBC-PLT position and direction provide an integrated trajectory-based risk metric. Two patient examples are provided, one involving recovery of an initially high-risk patient (**a**, Patient 1), and the other involving deterioration of an initially stable patient (**a**, Patient 2). See main text for case details. **b** Across the entire validation cohort, retrospective trajectory-based assessment identified patients during hospitalization at high risk of mortality and prolonged length of stay (LOS). Position and direction jointly contribute to risk, with patients outside the 80th percentile for both position and direction having elevated likelihood of poor outcomes. Compared to patients with good position and direction (<50th percentile), patients with poor position and direction (>80th percentile) on post-op day 4 had a 33-fold increased relative risk of adverse outcomes (CI: 13–79, 53% vs 1.6%) and stayed an average of 13 days longer in hospital.

**c** For other inflammatory stimuli studied, positional and directional deviation were similarly associated with significant increases in relative risks of adverse outcomes for patients with sufficient data available: 10x for adverse outcomes following hip replacement, 9x for colectomy, 9x for mortality from COVID-19, and 5x for mortality from myocardial infarction. See "Methods" for more details. See Supplementary Fig. 16 for risk stratification for other cohorts and Supplementary Figs. 17–22 and Supplementary Table 9 for position and direction and outcome likelihoods across the first 7 days post-op. Position and direction thresholds were calculated from the exploratory cohort, and outcome rates from the validation cohort. Position percentiles were calculated treating WBC below and PLT above the mean as non-contributing. Source data are provided as a Source data file. WBC: white blood cell count, PLT: platelet count.

## WBC-PLT trajectory-based patient risk assessment

Deviation from the direction of the favorable recovery trajectory shape is associated with elevated risk of adverse outcomes following cardiac surgery (Fig. 1d), and WBC-PLT position can also be considered to provide an integrated trajectory-based risk assessment. Illustrating this approach to individual patient risk assessment, the left panel of Fig. 4a plots the hospital course of a 55-year-old female with a prior mechanical aortic valve replacement and high STS pre-operative risk of mortality (PROM)[14,15] (10%, 95th percentile). Repeat sternotomy for aortic valve replacement was complicated by intra-operative bleeding and hypotension requiring rapid transfusion and resuscitation, but the post-op recovery was otherwise smooth. Daily comparison of the patient's WBC-PLT position and direction yields a high initial risk that steadily declines as WBC normalizes, and the trajectory merges quickly with the reference shape of recovery. In contrast, the right panel of Fig. 4a illustrates the clinical course for an 84-year-old female with a history of angina and moderate PROM (2.6%, 75th percentile) who initially recovered well after successful three-vessel coronary artery bypass grafting. Counts normalized by post-op day 4, but the rising WBC and subtly declining PLT thereafter nevertheless correspond to a

marked deviation from the favorable trajectory and a sharp rise in directional risk. On post-op day 6, she developed fever and shortness of breath leading to a diagnosis of pulmonary embolism and initiation of heparin. A precipitous interval decline in PLT on day 7 led to diagnosis of heparin-induced thrombocytopenia, intensive care unit readmission, and a subsequent prolonged hospital stay of 1 month. These two cases illustrate how the WBC-PLT recovery shape can be used to identify high-risk patients who recover smoothly, as well as to detect smoldering issues which precede adverse events. For the cardiac surgery cohort overall, patients outside the 80th percentiles of position and direction had a relative risk of adverse outcomes of 32 on day 3 (CI: 13–79, 1.4–44%) and 33 on day 4 (CI: 15–75, 1.6–53%). Extending this analysis to the other inflammatory scenarios studied revealed significant relative risks: 9.2 (2.7–31.2, 4.3–40%) for death from COVID-19, 4.9 (3.1–7.6, 5.8–28%) for death following myocardial infarction, and 10.1 (CI: 4–26, 5–50%) for hip replacement, with all results from out-of-sample testing (Fig. 4c). In the validation cardiac surgery cohort, 97% (29/30) of deaths occurred in patients whose day 4 position or direction was above the 80th percentile, with 83% (25/30) occurring in patients with both position and direction above the 80th

percentile. Corresponding specificities were 71% (865/1226) and 92% (1125/1226). Further risk analysis in the cardiac surgery cohort illustrated that consideration of the one-dimensional PLT/WBC ratio, or of PLT and WBC in isolation produced a significantly weaker risk stratification than the dynamic two-dimensional approach (Supplementary Table 4).

## Discussion

We identified a universal trajectory of human acute inflammatory recovery defined by exponential decay of WBC and delayed linear increase in PLT. This response program appeared in the setting of diverse traumatic, infectious, and ischemic inflammatory stimuli, irrespective of patient demographics and baseline risk, and was robust across multiple years of clinical care at two different hospitals. This WBC-PLT trajectory thus appears to represent an unrecognized fundamental core program of human physiology.

In retrospective analysis, deviation from this WBC-PLT recovery trajectory was associated with a 5–33x increased relative risk of adverse outcomes across 12 inflammatory cohorts and may help provide a generic personalized benchmark for tracking individual patient recovery, analogous to a pediatric growth-chart[18]. Complications due to additional inflammatory stimuli or other pathologic events may perturb the benchmark shape or superimpose other dynamics. The exhibited dynamics likely reflect the cumulative effect of more complex underlying phenotypes, such as modified blood cell production and clearance, as well as adaptive immune responses within the leukocyte population[19]. Recognition of this WBC-PLT pattern may also help guide investigation of the molecular and cellular signals driving functional and dysfunctional inflammatory responses. The robustness of this recovery shape among recovering patients shows that healthy inflammatory responses are similar, in contrast to those associated with adverse outcomes (Fig. 1, Supplementary Figs. 4–5) which can be highly divergent. In general, this contrasting homogeneity and heterogeneity for successful and unsuccessful outcomes has been identified in other biological systems as the Anna Karenina Principle[20].

The mathematical characterization of inflammatory recovery as exponential WBC decay toward baseline with delayed linear PLT growth toward baseline is consistent with conceptual models of homeostatic control of WBC and PLT relative to their setpoints[1,4,21]. The rate of PLT growth (modeled by $k_{PLT_{growth}}$) varied significantly with inflammatory stimulus, but the rate of WBC decline ($k_{WBC_{decay}}$) varies less, despite large variation in WBC (Supplementary Fig. 7). The range of $k_{WBC_{decay}}$ (0.5–1.0 days$^{-1}$) is similar to experimentally observed neutrophil clearance rates[22] and monocyte clearance times in myocardial infarction[23]. The delay in PLT growth starting ~1–2 days after peak WBC is consistent with the hypothesis that hemostasis has been achieved shortly after pro-inflammatory signals have peaked. Recoveries for many inflammatory conditions appear to overshoot, reaching $PLT(t) > PLT_{setpoint}$ (Fig. 2). Future investigations are needed to validate this observation and more generally to define the logic of the hypothesized control systems, dissecting the contributions of production, consumption, senescence, and other component processes. Future work is also needed to understand how each patient's pre-operative WBC-PLT setpoint is stored to enable consistent recovery toward that baseline.

The WBC-PLT trajectory provides one possible unifying explanation for a number of prior studies reporting associations between ratios of different CBC indices (e.g., WBC/PLT, PLT/neutrophil, etc.) and clinical outcomes in a range of specific disease states[24–30]. During the WBC-PLT recovery trajectory, WBC count decreases, and PLT counts increases, meaning that a ratio such as WBC/PLT will necessarily decrease. The prognostic associations identified for these ratios likely reflect some of the same physiologic processes generating the WBC-PLT recovery trajectory. Since a simple ratio cannot adjust for differences in baseline setpoints or time delays for each patient, and

since rates of change in the ratio may not reflect patient-specific variation in the WBC decay rate and the PLT growth rate, it is expected that these ratios show less prognostic association than the two-dimensional WBC-PLT trajectory (Supplementary Table 4).

Leukocytosis is a cardinal sign of inflammation, but a small fraction of patients with the inflammatory conditions studied do not show elevation in WBC, and some can at times show a decrease. In the cardiac surgery cohort -0.5% (25/4693) of patients failed to show a WBC increase in the 72 h following surgery; in the MI cohort, 5% (314/6240) had at least one low WBC count during the 72 h following admission; and in sepsis—where timing of peak inflammation may be hard to ascertain—89% (4209/4730) of patients either exhibited leukocytosis, or a sharp (>1 unit) WBC increase within a 24 h period at some point during their stay. Future study is needed to investigate whether these atypical muted or decreasing WBC responses are related to treatment or intrinsic to the inflammatory response and whether inflammatory recovery follows a different trajectory in the case of transient leukopenia.

All major results were validated in out-of-sample testing or using cross-validation approaches. Results for the cardiac surgery cohort were replicated in separate cohorts both from the same hospital during different time periods and from a different hospital in the same network, and in subset analyses controlling for surgery type, pre-operative marker levels, and risk profiles. However, this study was limited to retrospective analysis, and some patient cohorts were restricted by the required minimum data availability. Follow-up prospective studies with higher-frequency measurements are needed to define these responses more precisely and to determine whether interventions guided by awareness of this conserved recovery shape will lead to improved patient outcomes. Future work is also warranted to determine whether the shape of the effective inflammatory response identified here is present in other inflammatory settings such as malignancy, autoimmunity, and the resolution of chronic inflammation.

## Methods

The study was performed in compliance with a research protocol approved by the local institutional review board (IRB). The Mass General Brigham IRB approved access to patient identifiers for the purpose of linking different sources of patient data during the analysis phase of the project, with links to patient identifiers to be destroyed at the conclusion of the project. The IRB approved publication of only summary patient data or limited individual de-identified patient data. The IRB considered the risks to research subjects, the selection of subjects, the privacy of subjects, and confidentiality of the data and determined that the study's data management procedures and publication plans justified an authorization for a waiver of informed consent.

### Cardiac surgery cohort

Data for the cardiac surgery cohort was collected from all adult Massachusetts General Hospital (MGH) patients who underwent cardiac surgery between 01-01-2016 and 31-12-2019 ($N = 6054$). All blood count (CBC) and basic metabolic panel (BMP) measurements during the surgery-associated hospital stay were collected from the Partners Healthcare Electronic Data Warehouse (EDW) and Research Patient Data Registry (RPDR). Patient demographics, risk factors, surgical factors, and post-operative outcomes were collected from a dataset adjudicated by the MGH Division of Cardiac Surgery for contribution to the national Society of Thoracic Surgeons (STS) database. Patients were excluded if their surgery was an emergency or salvage procedure, if they were discharged within 48 h of surgery, or if they did not have at least 1 valid post-op CBC and BMP (post-exclusion $N = 4693$). Analysis was initially limited to surgeries between 01-01-2016 and 30-09-2018 ($N = 3168$, exploratory cohort). All key findings were then validated

using surgeries from 01-10-2018 to 31-21-2019 ($N = 1525$, validation cohort). An independent cohort was derived from cardiac surgeries at Brigham and Women's Hospital (BWH) between Jan-01-2016 and Sep-30-2019 ($N = 1988$). Reference intervals for blood count and metabolic panel biomarkers at both MGH and BWH are given in Supplementary Table 5.

## Other inflammatory cohorts

Following analysis of the cardiac cohort, clinical data were collected from all adult MGH patients who underwent any of 6 major surgeries (colectomy, Cesarean section, hip arthroplasty, hysterectomy, limb amputation, Whipple procedure), or had any of 5 acute inflammatory diagnoses (myocardial infarction [MI], stroke, *C. difficile* colitis infection, or novel coronavirus infection [COVID-19], and sepsis), between 01-01-2016 and 31-12-2020. Patients discharged within 48 h, without at least 2 CBCs, or those undergoing laparoscopic procedures were excluded, as were repeat visits from the same patient for the same event type. All stay-associated CBCs and BMPs were collected, along with demographics, admission and discharge dates, and all-cause mortality, using RPDR and EDW.

For all cohorts, demographics, outcomes, and all hematology (e.g., CBC) and clinical chemistry (e.g., BMP) test panels measured during the associated admission were retrieved. Analysis included all lab tests that were routinely collected at least once per day for hospitalized patients at the study hospitals (20 total measurements listed in Supplementary Table 5). Adverse outcomes were defined as all-cause mortality within 30 days of discharge and long length of hospital stay (LOS) relative to typical cohort LOS: LOS >10 for surgeries with mean LOS < 7 (Cesarean section, hip arthroplasty, hysterectomy) and LOS >14 otherwise. Cohort summary statistics are given in Table 1.

## Cohort exclusion criteria

Patients were excluded if they were under 18 yrs old, had a <2 day associated hospital stay, or underwent a laparoscopic procedure (for surgical cohorts). Some cohorts had additional inclusion/exclusion criteria specific to that setting:

- Amputation: Surgeries were included only if they involved amputation of a leg (above or below knee), arm (above or below elbow), or of a whole foot or hand. Amputations of fingers or toes were not included.
- Colectomy: Colectomy was defined as any major (invasive, non-laparoscopic) surgery of the small or large bowel and was predominantly small bowel resection or full or partial large bowel resection/colectomy.
- Stroke: Stroke was defined as any diagnosis of a stroke or cerebrovascular accident.
- *C. difficile* colitis: The colitis cohort was limited to patients with a diagnosis of *C. difficile* colitis or infectious colitis, or with a diagnosis of colitis, and a confirmed positive *C. difficile* toxin assay.
- Sepsis: The sepsis cohort included any patients with diagnosis of sepsis regardless of whether the underlying infection/organism was specified and encompassed both mild and severe sepsis diagnoses.

Following exclusions, each cohort was split evenly, with the earlier half (by diagnosis or surgery date) taken as the exploratory set, and the latter half taken as the validation set. Cohort sizes after each exclusion are show in Supplementary Table 6.

The cardiac surgery cohort was derived from a manually curated dataset adjudicated by the Massachusetts General Hospital Division of Cardiac Surgery for contribution to the national Society of Thoracic Surgeons (STS) database. For all other cohorts, data was collected by filtering electronic health record databases for keywords associated with the surgery or diagnosis. For each cohort, terms were selected based on author clinical experience. For each cohort, a random sample of patient health records was manually checked to ensure that the database-listed diagnosis and procedures accurately reflected information in patient medical records. Due to the nature of the Partners Healthcare network databases, a small number of patients in each (non-cardiac surgery) cohort may not have received treatment exclusively at MGH, instead receiving part of their treatment at one of the other hospitals in the Partners Healthcare network.

In addition to total cohort exclusions above, for accurate calculation of trajectories and model parameter estimation in Fig. 2, patients with 5 days of post-WBC peak data or more were included. To focus on situations where biological variation could be confidently distinguished from analytic variation, patients who did not have an inflammation-induced WBC increase of at least 2 units were excluded.

## Dimensionality reduction and trajectory calculation

For preliminary analysis of inflammatory trajectories, clinical laboratory test results were interpolated and evenly sampled every 12 h post-operation. Patient response clusters in the cardiac exploratory cohort (Supplementary Fig. 1) were derived using k-means clustering[31] applied to patient lab test measurements throughout the surgery-associated hospital stay. Measurements were normalized (by pre-operative means), interpolated, and sampled every 12 h until discharge, with post-discharge values set to 0. The number of clusters (5) was the maximum number which resulted in all groups having more than 50 patients. Patients with fewer than 3 sets of measurements were not included when defining clusters but were assigned to their nearest group afterward. Clinical tests which showed insignificant (<10%) variation across clusters were excluded, leaving 10 measurements. Given the high correlation (Pearson coefficient >0.9) between HCT, HGB, and RBC, only HCT was included in the final clustering, leaving 8 measurements: anion gap (ANION), blood-urea nitrogen (BUN), creatinine (CRE), hematocrit (HCT), glucose (GLU), platelet count (PLT), red cell distribution width (RDW), and WBC. Statistical significance of trajectory clusters was tested through comparison to synthetic data, using a method outlined by Liu et al.[31].

Dimensionality was further reduced by identifying the optimal pair of results (from the 28 possible combinations) for risk stratification. High-performance pairs were identified by considering the significance ($p$-value, $\chi^2$ test) of a generalized linear model predicting mortality using the interpolated post-op day 1 measurements from each biomarker pair. Clinical and physiologic interpretability were also subjectively considered, validating the choice of WBC-PLT for 2D distillation of the high-dimensional dataset.

## WBC-PLT phase plane analysis

In the exploratory cardiac surgery cohort, dynamics in the WBC-PLT phase plane were analyzed by calculating the mean WBC-PLT response for patients who survived with hospital stays of <14 days. Mean responses were then calculated for subsets of this cohort stratified by age, length of hospital stay, medical center, pre-op lab values, pre-op risk evaluations, and surgery type (Fig. 1c). Preliminary risk evaluations (Fig. 1d) were calculated by comparing the angle of deviation of validation cohort patients against the reference trajectory to the associated deviation distribution in the exploratory cohort. Angles against the reference trajectory were calculated using the WBC and PLT change over the past 24 h, after normalization by their pre-operative means.

## Mathematical modeling

Models of exponential decay and linear growth were fit to WBC and PLT counts. To account for differences in precise timing of the initiation of the acute inflammatory response, patients were aligned according to the time of their peak WBC count: in the 3 days after surgery (surgery cohorts), in the 3 days after admission (ischemia), or

over their entire stay (infection). An exponential decay model ($y = a + be^{-ct}$; parameters: a, b, c) and a linear model ($y = d + f \cdot t$; parameters: d, f) were fit to patient WBC and PLT counts over the first 5 days after peak WBC. The PLT model was fit after a lag $\tau$, which allows for potential delay between peak WBC and the start of PLT growth. The favorable response for each cohort (Fig. 2) was calculated as the mean WBC-PLT trajectory after peak WBC for all patients without adverse outcomes who had at least 5 days of WBC and PLT measurements after peak WBC and whose peak WBC was at least $2 \cdot \frac{10^3 \text{cells}}{\mu l}$ above their initial WBC (see Supplementary Table 6 for cohort sizes). Example code for trajectory calculation and model fitting is provided in Supplementary Software.

**WBC-PLT trajectory-based risk analysis.** Patient risk of adverse outcomes was calculated for an independent validation cohort of patients by comparing a patient's position and movement direction in the WBC-PLT phase-plane to the mean WBC-PLT trajectory calculated for patients in an independent exploratory cohort without adverse outcomes. Positional risk was calculated as the distance from the mean trajectory, after normalizing WBC and PLT by baseline means and treating WBC below and PLT above the mean as contributing zero distance. Directional risk was calculated as the angle between the patient's normalized daily WBC and PLT change vector and the normalized mean WBC-PLT trajectory change vector. Each patient's relative position and direction were converted to percentiles relative to distributions in exploratory cohorts. All thresholds were determined using the exploratory cohorts, and all risk stratifications based on these thresholds were calculated for the validation cohorts. Choices of percentile thresholds in Fig. 3 were made from analysis of only the cardiac surgery exploratory cohort. Because of uncertain inflammatory event timing in the infection cohorts, risk calculations were performed after patient alignment using the timing of peak WBC count within the first 72 h after admission. Risk stratification remained significant regardless of this alignment (Supplementary Fig. 11).

Risk analysis in the cardiac surgery cohort was also performed using other high-significance test result pairs (WBC-RDW, WBC-BUN, PLT-ANION), with WBC-PLT maintaining the strongest risk stratification (Supplementary Tables 2–3) and illustrating a more robust recovery trajectory (Supplementary Fig. 12). Risk stratification in the cardiac surgery cohort was also preserved when performed using non-interpolated laboratory values (Supplementary Tables 7–8), with patient position and direction at a given time calculated using the last two available blood counts (excluding repeats). Example code for trajectory calculation and risk analysis is provided in Supplementary Software.

**Statistical analysis.** Statistical analysis was performed in *MATLAB*. For all continuous variables, unless otherwise noted, we report means and use analysis of variance (ANOVA) for population comparisons. For categorical variables we report percentages and use a chi-square test for population comparisons. Thresholds for statistical significance were set at $p = 0.05$. For event rates, confidence intervals were calculated assuming binomial distributions. WBC and PLT trajectories were interpolated and evenly sampled every 12 h during patient admission.

#### Reporting summary
Further information on research design is available in the Nature Research Reporting Summary linked to this article.

## Data availability
Source data for all figures are provided with this paper in Supplementary Information. IRB restrictions prevent sharing of some of the detailed raw data analyzed in this study. Consistent with this requirement, a sample dataset has been included in Supplementary Information, along with custom code to illustrate how the primary figures

and results are generated. Any additional data not included in the source data or sample dataset and not restricted by IRB approval is available upon request from the corresponding author (john_higgins@hms.harvard.edu). Source data are provided with this paper.

## Code availability
Custom code for calculating WBC-PLT trajectories (Fig. 1), fitting WBC and PLT models (Figs. 2–3), and for calculating position-direction-based risk (Fig. 4) are provided in Supplementary Software. An anonymized dataset compatible with IRB approval has been provided to demonstrate the use of the code.

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

## Acknowledgements

We thank Rahul Deo and Erik Reinertsen for helpful feedback, Chin Siang Ong and Steven Song for help with access and orientation to datasets, David Louis, Rebecca Ward, and Sunny Dzik for helpful discussions, and the Partners Healthcare Research Patient Data Registry and Electronic Data Warehouse groups for facilitating use of their database. Portions of the computational analysis in this study were conducted on the O2 High Performance Compute Cluster at Harvard Medical School. This work was supported by the One Brave Idea Initiative (J.M.H.), Fast Grants at the Mercatus Center, George Mason University (J.M.H.), NIH grant DP2DK098087 (J.M.H.), the Partnership for Clean Competition Research Collaborative (J.M.H.), the MGH Hassenfeld award (A.D.A.), the Controlled Risk Insurance Company/Risk Management Foundation, CRICO (A.D.A.).

## Author contributions

A.D.A., B.H.F., J.C.T.C., and J.M.H. designed the study, analyzed data, and wrote the manuscript. A.D.A., B.H.F., J.C.T.C., J.M.H., and T.M.S. contributed to data collection, data interpretation, and manuscript editing. B.H.F. wrote custom code.

## Competing interests

A.D.A., B.H.F., J.C.T.C., and J.M.H. are named as inventors on a patent application related to this work submitted by Mass General Brigham. No authors have any other competing interests.
