## [Peer Review File · Nature Communications]

Human Acute Inflammatory Recovery is Defined by Co-Regulatory Dynamics of White Blood Cell and Platelet PopulationsPeer review file, first round review comments –

Reviewer #1 (Remarks to the Author):

I think the authors have adequately addressed previous referees' comments and provided additional explanations and clarifications in the revised manuscript. I certainly see the value of this approach, which is currently under-developed in the field. As such, this study would be valuable to a broad readership and will likely promote follow up studies into underlying molecular mechanisms, clinical utility, as well as additional biomarker discovery based on the findings presented here.

I therefore recommend publication of this study in Nat Comm.

Reviewer #2 (Remarks to the Author):

Foy et al present a retrospective, clinical observational study describing the potential utility of dynamic assessment of white blood cell (WBC) to platelet (PLT) ratio as a means of prognosticating adverse outcomes in the context of acute inflammation in various contexts. There are multiple flaws in this study:

1. While the authors are to be commended on having studied a large series of patients, the studies are unfortunately not novel. A cursory search of PubMed yields numerous prior studies in this regard (none of which have been cited by the authors):

Prior studies documenting WBC to PLT ratio as being prognostic:

<https://pubmed.ncbi.nlm.nih.gov/31481452/>; dynamic changes in these ratios

(<https://pubmed.ncbi.nlm.nih.gov/27911876/>; <https://pubmed.ncbi.nlm.nih.gov/30341514/>);

multiple

variants thereof in other disease settings: <https://pubmed.ncbi.nlm.nih.gov/30809980/>;

<https://pubmed.ncbi.nlm.nih.gov/31282096/>; <https://pubmed.ncbi.nlm.nih.gov/32369297/>;

<https://pubmed.ncbi.nlm.nih.gov/30116146/>; <https://pubmed.ncbi.nlm.nih.gov/26403379/>);

<https://pubmed.ncbi.nlm.nih.gov/29715157/>; many more citations are not listed.

2. Another central flaw is that while the authors state repeatedly the elevations in WBC are a hallmark of inflammation and critical illness, the neglect to mention that the opposite (a depression in WBC counts) is also a hallmark of some patients suffering from, for example, sepsis. Furthermore, severe trauma (such as that studied in the present study) induces an increase in innate immune gene expression concomitantly with reduced adaptive immune gene expression in circulating leukocytes, suggesting a much more complex phenotype than simple elevation in WBC. This is likely a central reason that the prognostic utility of WBC in some of these rapidly evolving disease states has been limited.

3. Other salient issues are methodological. The authors report selecting 20 biomarkers out of the large number available to them, but do not report what they were or why only these 20 were selected. The state that WBC and PLT are highly cross-correlated; if so, it would seem that any statistical outcome prediction models would exclude one or the other variable rather than using both, as was done in this study. They use remarkably simplistic statistical model structures but do not report the actual models used (i.e. with all parameters stated explicitly).

Reviewer #3 (Remarks to the Author):

The authors responded adequately to the comments of Reviewer 3

Response to reviews of

"Human Acute Inflammatory Recovery is Defined by Co-Regulatory Dynamics of White Blood Cell and Platelet Populations"

REVIEWER COMMENTS

Reviewer #1 (Remarks to the Author):

I think the authors have adequately addressed previous referees' comments and provided additional explanations and clarifications in the revised manuscript. I certainly see the value of this approach, which is currently under-developed in the field. As such, this study would be valuable to a broad readership and will likely promote follow up studies into underlying molecular mechanisms, clinical utility, as well as additional biomarker discovery based on the findings presented here. I therefore recommend publication of this study in Nat Comm.

We greatly appreciate the supportive comments, and we share the reviewer's opinion that our study provides a roadmap for many important follow-up investigations to define molecular mechanisms, identify scenarios with greatest clinical utility, and develop even more powerful biomarkers.

Reviewer #2 (Remarks to the Author):

Foy et al present a retrospective, clinical observational study describing the potential utility of dynamic assessment of white blood cell (WBC) to platelet (PLT) ratio as a means of prognosticating adverse outcomes in the context of acute inflammation in various contexts.

We appreciate Reviewer #2's consideration of our manuscript and the detailed suggestions below that have helped us greatly improve the clarity of study. We first note that the "white blood cell (WBC) to platelet (PLT) **ratio**," is not exactly what we studying, and we appreciate this feedback and the opportunity to revise our manuscript to make clear the major conceptual differences between a static ratio and multivariate dynamics, with further details below.

There are multiple flaws in this study:

- 1. While the authors are to be commended on having studied a large series of patients, the studies are unfortunately not novel. A cursory search of PubMed yields numerous prior studies in this regard (none of which have been cited by the authors):** Prior studies documenting WBC to PLT ratio as being prognostic:
<https://pubmed.ncbi.nlm.nih.gov/31481452/>; dynamic changes in these ratios (<https://pubmed.ncbi.nlm.nih.gov/27911876/>; <https://pubmed.ncbi.nlm.nih.gov/30341514/>); multiple variants thereof in other disease settings:
<https://pubmed.ncbi.nlm.nih.gov/30809980/>;
<https://pubmed.ncbi.nlm.nih.gov/31282096/>; <https://pubmed.ncbi.nlm.nih.gov/32369297/>;

<https://pubmed.ncbi.nlm.nih.gov/30116146/>; <https://pubmed.ncbi.nlm.nih.gov/26403379/>); <https://pubmed.ncbi.nlm.nih.gov/29715157/>; many more citations are not listed.

We thank the reviewer for providing citations to studies of ratios of some CBC indices and their association with clinical outcomes in some specific disease contexts. We now make more clear in our revised Results and Discussion that we are not studying a ratio (a one-dimensional scalar quantity) as all the provided citations do. Our study is therefore novel in this way and others. We take a novel approach of studying multivariate (two-dimensional) dynamics and their dependence on time. None of the references provided above nor any other that we are aware of studies any two-dimensional trajectory during human inflammatory response, nor does any of the cited studies develop any mathematical model of the time-dependence of even those one-dimensional ratios, let alone the two-dimensional trajectory we study. Because these previous studies limit their analysis to a one-dimensional ratio, none of them is able to identify the unified inflammatory response we find across a wide range of disease states. Furthermore, our discovery of this WBC-PLT recovery trajectory and its broad association with disease states provides a unifying mechanistic explanation for prognostic association reported for the many different ratios described in the citations the reviewer provides, as we note in the Discussion of our revised manuscript and the new Supplemental Table 4.

2. Another central flaw is that while the authors state repeatedly the elevations in WBC are a hallmark of inflammation and critical illness, the neglect to mention that the opposite (a depression in WBC counts) is also a hallmark of some patients suffering from, for example, sepsis.

The reviewer notes that some patients with sepsis and related conditions can have depressed WBC counts instead of elevated, and we have revised our manuscript to consider these patients, including in the 4730-patient sepsis cohort we study where fewer than 1% of patients exhibit a depressed WBC count. This subset is interesting for follow-up study, as we note in our revised manuscript, but its small size and representation means that the inflammatory cohorts overall consistently follow the same canonical WBC-PLT recovery trajectory.

Furthermore, severe trauma (such as that studied in the present study) induces an increase in innate immune gene expression concomitantly with reduced adaptive immune gene expression in circulating leukocytes, suggesting a much more complex phenotype than simple elevation in WBC. This is likely a central reason that the prognostic utility of WBC in some of these rapidly evolving disease states has been limited.

We agree with the reviewer that severe trauma and other inflammatory stimuli trigger complex and heterogeneous responses, and as we discuss in our revised manuscript, this heterogeneity does currently limit the prognostic utility of simple WBC counts. We also emphasize in our revised Result and Discussion that this observation further underscores the impressive prognostic utility demonstrated in our study, suggesting that the prognostic utility of WBC counts alone, when understood and interpreted in the context the WBC-PLT trajectory, can be substantial: for instance identifying patients with >33x increased risk of adverse outcomes.

3. Other salient issues are methodological. The authors report selecting 20 biomarkers out of the large number available to them, but do not report what they were or why only these 20 were selected.

We have expanded the description of our Methods in our revised manuscript, including moving additional detail from the supplementary materials to the main Methods section in order to make clear that we studied all of the standard protocol clinical laboratory tests that are routinely ordered for hospitalized patients at our study hospitals, and we provide a complete list of these tests in Supplementary Table 9.

The state that WBC and PLT are highly cross-correlated; if so, it would seem that any statistical outcome prediction models would exclude one or the other variable rather than using both, as was done in this study.

Correlation coefficients for pairs of lab tests reached as high as ~ 0.4 , and as we note in our revised manuscript, while this level of correlation is significant and provides constraints for mechanistic models, it does not approach the level of redundancy (correlation coefficient near 1.0) that would be required to allow complete exclusion of one variable for purposes of prognostication. We illustrate this point directly by comparing prognostic associations of the individual variables in a new Supplementary Table S9.

They use remarkably simplistic statistical model structures but do not report the actual models used (i.e. with all parameters stated explicitly).

Our revised manuscript includes descriptions of all statistical models and parameters in the first two Results sections, Figure 1b, and Supplementary Tables 2-3, and we moved some supplementary methods descriptions to the revised main Methods section to provide additional clarity. We have also now provided Supplementary Software to show how all statistical tests were performed and how figures were generated. The “remarkable simplicity” of our models noted by the Referee is indeed a strength of our study which succeeds in describing the dynamics of inflammatory recovery from a broad range of pathologic conditions with a simple model that can be easily understood and integrated with existing physiologic understanding and clinical intuition.

Reviewer #3 (Remarks to the Author):

The authors responded adequately to the comments of Reviewer 3

We thank the reviewer for evaluating our manuscript.

Peer review file, second round review comments –

Reviewer #2 (Remarks to the Author):

The reviewers have revised their manuscript, and this addresses my salient concerns.

Response to reviews of revised version of
*"Human Acute Inflammatory Recovery is Defined by Co-Regulatory
Dynamics of White Blood Cell and Platelet Populations"*

REVIEWERS' COMMENTS

Reviewer #2 (Remarks to the Author):

The reviewers have revised their manuscript, and this addresses my salient concerns.

We appreciate Reviewer #2's careful consideration of our revised manuscript.